# Chromium-catalyzed *para*-selective formation of quaternary carbon centers by alkylation of benzamide derivatives

Pei Liu[1,2,3], Changpeng Chen[1,3], Xuefeng Cong[1,3], Jinghua Tang[1,3] & Xiaoming Zeng[1,3]

Selective creation of quaternary carbon centers has been a long-standing challenge in synthetic chemistry. We report here the chromium-catalyzed, *para*-selective formation of arylated quaternary carbon centers by alkylative reactions of benzamide derivatives with tertiary alkylmagnesium bromides at room temperature. The reaction, which was enabled by a low-cost chromium(III) salt combined with trimethylsilyl bromide, introduces a sterically bulky tertiary alkyl scaffold on the *para*-position of benzamide derivatives in a highly selective fashion without either isomerization of the tertiary alkyl group or formation of *ortho*-alkylated byproducts. Forming low-valent Cr species in situ by reaction of CrCl$_3$ with t-BuMgBr accompanied by evolution of hydrogen can be considered, which serves as reactive species to promote the reaction. The *para*-alkylation likely occurs via a radical-type nucleophilic substitution of imino-coordination benzimidate intermediate.

[1] Key Laboratory of Green Chemistry and Technology, Ministry of Education, College of Chemistry, Sichuan University, Chengdu 610064, China. [2] Department of Applied Chemistry, School of Science, Northwestern Polytechnical University, Xi'an 710072, China. [3] Frontier Institute of Science and Technology, Xi'an Jiaotong University, Xi'an 710054, China. Correspondence and requests for materials should be addressed to X.Z. (email: zengxiaoming@scu.edu.cn)

Transition-metal-catalyzed alkylative reactions are fundamental transformations in synthetic chemistry, and they represent a powerful tool with which to incorporate aliphatic scaffolds into molecules; such reactions have been used for the construction of pharmaceuticals and materials[1–3]. However, the introduction of bulky tertiary alkyl groups into motifs for the catalytic formation of quaternary centers has long been a prominent challenge because of the effect of steric hindrance, competing β-hydride elimination and the ease with which such moieties undergo isomerization[4–13]. To create arylated quaternary carbon centers, aryl halides, triflates, and organoborons are usually used as aromatic sources to react with tertiary alkyl nucleophiles or electrophiles. These approaches were pioneered by Biscoe[14], Fu[15], Gong[16], and others[17–19] and typically employed nickel catalysis (Fig. 1a). In contrast, the use of aromatic hydrocarbons for the catalytic formation of arylated quaternary carbon centers has rarely been studied.

Given that aromatic hydrocarbons usually contain site-differentiated C–H bonds, regioselectivity in the incorporation of bulky tertiary alkyl groups is a formidable obstacle. The application of common methods involving ortho-alkylation to introduce arylated quaternary carbon centers may be challenging because of the effect of steric repulsion[20]. A landmark study by Nakao showed that arylated 2° carbon centers can be created in a para-selective manner with benzamides through nickel/aluminum co-catalyzed hydroarylation (Fig. 1b)[21–23]. We questioned whether it was possible to construct quaternary carbon centers at the para-position of benzamides by using a transition-metal-catalyzed alkylative reaction with sterically bulky tertiary alkyl nucleophiles.

Recently, transition-metal catalysis using abundant, low-cost base metals such as nickel, iron, and cobalt has appeared as a cost-effective tool for organic synthesis[24–29]. In contrast to the great achievements made with these first-row metals, synthetic chemistry of Group 6 metal chromium has still been underdeveloped[30–40]. Herein, we report that the para-selective formation of arylated quaternary carbon centers was enabled by using low-cost chromium(III) salt combined with trimethylsilyl bromide to achieve alkylative reaction of benzamides with tertiary alkylmagnesium bromides at room temperature (Fig. 1c). This reaction proceeded with high selectivity, with only the para-carbons of benzamides being alkylated without isomerization of bulky tertiary alkyl groups.

## Results

### Reaction optimization

Based on our previous results, the treatment of chromium salt with PhMgBr allowed the formation of low-valent species, which show high catalytic activity in the selective cleavage of inert C–O and C–N bonds[34,35]. We initially probed the reactivity of chromium in promoting the alkylation of N-methylbenzamide (1) with t-butylmagnesium bromide (Table 1). With 10 mol% CrCl3, the alkylation of benzamide with t-BuMgBr did not occur (entry 1). Conducting the reaction with 2,3-dichlorobutane (2,3-DCB) as the additive did not give the alkylated compound (entry 2). Gratifyingly, chlorodimethyl (phenyl)silane could be used to promote the Cr-catalyzed alkylation, in which the bulky t-butyl group was selectively incorporated at the para-position of benzamide to afford the compound 3 in 44% yield (entry 3). In contrast, the alkylation did not take place using dichlorodiphenylsilane (entry 4). The inclusion of bromotrimethylsilane (TMSBr) greatly improved the transformation, giving 3 in preparatively useful yield (entry 6). The use of other chromium salts such as CrCl2 and Cr(acac)3 led to inferior result (entries 7 and 8). Other first-row transition-metal complexes such as Ni(cod)2 and CoCl2 were completely inactive in

promoting the para-selective alkylation using t-BuMgBr (entries 9 and 10). Interestingly, the para-alkylation with 10 mol% of FeCl3, AlCl3, or AlMe3 also occurred, albeit with low conversions (entries 11–13).

### Substrate scope

The substrate scope of the para-alkylation was then examined for the construction of structurally diverse substituted quaternary carbon centers with benzamides. As shown in Fig. 2, the benzamide derivative containing an ortho-methyl, benzyl, phenoxymethyl, or tert-butoxy(phenyl)methyl substituent reacted with t-BuMgBr smoothly to form the para-alkylated compound (4–6). Alkylation using benzamide bearing an electron-withdrawing fluoride substituent on the ortho-position gave an inferior result compared with those bearing electron-donating groups (7 and 8). Meanwhile, the incorporation of alkoxy and phenoxy groups into the ortho position of benzamides did not affect the para-selective alkylation of C–H bonds (8–11). Interestingly, the Cr-catalyzed reaction of 2-hydroxy-N-methyl-benzamide occurred smoothly to give 2-hydoxyl and 4-tert-butyl-substituted benzamide derivative, albeit with a low conversion (12). The para-C–H bonds in the scaffolds of N-methylbenzamides bearing an ortho-methylthio or trimethylsilyl group can be effectively alkylated under present conditions, providing access to the desired products 13 and 14 in 61% and 52% yields, respectively. We were pleased to find that [1,1'-biphenyl]-2-carboxamide motifs containing functional substituents of methyl, phenyl, chloride, trifluoromethyl, methylthio, amino, and alkoxycarbonyl groups could couple with tert-butyl Grignard at the amide-bearing aromatics, the formation of diverse-substituted derivatives 15–23 in preparatively useful yields. In addition, ortho-naphthyl and thienyl-bearing benzamides were also amenable to the para-alkylative cross-coupling reaction (24 and 25). It was noteworthy that steric hindrance arising from the meta-substituents of benzamides did not affect the site-selectivity of alkylation, allowing for incorporating the bulky tert-butyl group at the para-position of benzamides (26–28). A broad range of functionalities, such as chloride, methylthio, trifluoromethyl, trifluoromethoxy, trimethylsilyl, hydroxyl, amino, alkoxycarbonyl, naphthyl, and thienyl were well retained under the reaction conditions. Importantly, this Cr-catalyzed para-selective alkylation can be applied to prepare tri-substituted N-methyl-benzamide derivative that contains 2-fluoro-3-methoxy, 2,3-dimethoxy or dihydrobenzo[b][1,4]dioxine scaffold (29–31). Interestingly, the alkylation using N-methylthiophene-2-carboxamide led to the formation of a quaternary carbon center at the C5 position of thiophene, leading to 5-tert-butyl-substituted thiophene derivative 32. The chromium-catalyzed protocol is scalable, and can be applied to the synthesis of para-alkylated benzamide 3 on a gram scale.

In addition to t-butylmagnesium bromide, tertiary alkyl nucleophiles, such as 2-methyl-4-phenylbutan-2-yl, 2-methyl-hexan-2-yl, 2-methylnonan-2-yl, t-pentyl, 3-ethylpentan-3-yl, and methylcyclohexyl-substituted Grignard reagents also reacted with N-methylbenzamide smoothly under chromium catalysis, permitting the incorporation of sterically bulky tertiary alkyl scaffolds into the para-position of benzamide in the synthesis of related compounds 33–38 (Fig. 3). However, the para-alkylative reaction with adamantyl-substituted Grignard reagent furnished the coupling product 39 in low yield. Variation of N-methyl group to ethyl and substituted phenyl in the benzamide motifs did not hamper the para-selective transformation (40–43). Notably, these Cr-catalyzed alkylation reactions all proceeded with high selectivity without the formation of tertiary alkyl isomerized side products, and only the para-carbons of benzamides were alkylated at room temperature. However, the

**a Prior work:** Ni-catalyzed creation of arylated quaternary carbon centers

| X | Y | Catalyst | Reaction type |
|---|---|---|---|
| [Mg] | Br, OTf | [Ni] | Kumada coupling [Biscoe, Glorius] |
| Halide, OR | [B] | [Ni] | Suzuki coupling [Fu, Watson, Molander] |
| Halide | Br | [Ni] | Reductive coupling [Gong] |

**b Prior work:** *para*-selective formation of C–C bonds with benzamides [Nakao]

**c This work:** Cr-catalyzed *para*-selective formation of quaternary carbon centers with benzamide derivatives

*para*-selectivity
Cost-effective, low-valent Cr catalysis
No isomerization of tertiary alkyl groups
Without forming *ortho*-alkylated byproducts

Forming quaternary carbon center

**Fig. 1** Transition-metal-catalyzed formation of arylated quaternary carbon centers by alkylation. **a** Known examples of the formation of arylated quaternary carbon centers with nickel catalysis. **b** *para*-Selective alkylation of benzamide with nickel catalysis. **c** Cr-catalyzed *para*-alkylation of benzamides for the formation of arylated quaternary carbon centers

---

### Table 1 Optimizing reaction conditions[a]

| Entry | Metal salt | Additive | 3 (%) |
|---|---|---|---|
| 1 | $CrCl_3$ | — | ND |
| 2 | $CrCl_3$ | 2,3-DCB | ND |
| 3 | $CrCl_3$ | $PhMe_2SiCl$ | 44 |
| 4 | $CrCl_3$ | $Ph_2SiCl_2$ | ND |
| 5 | $CrCl_3$ | TMSCl | 49 |
| 6 | $CrCl_3$ | TMSBr | 77 (72)[b] |
| 7 | $CrCl_2$ | TMSBr | 48 |
| 8 | $Cr(acac)_3$ | TMSBr | 53 |
| 9 | $Ni(cod)_2$ | TMSBr | ND |
| 10 | $CoCl_2$ | TMSBr | ND |
| 11 | $FeCl_3$ | TMSBr | 8 |
| 12 | $AlCl_3$ | TMSBr | 14 |
| 13 | $AlMe_3$ | TMSBr | 12 |

GC yields were given using *n*-tridecane as internal standard
[a]Conditions: **1** (0.2 mmol), **2** (0.8 mmol), metal salt (0.02 mmol), additive (0.6 mmol), THF, rt, 24 h
[b]Isolated yield in parenthesis
ND, not detected

**Fig. 2** Cr-catalyzed *para*-selective alkylation of benzamides with *tert*-butyl Grignard reagent. Conditions: *N*-methylbenzamide derivative (0.2 mmol), *tert*-butylmagnesium bromide (0.8 mmol), CrCl₃ (0.02 mmol), TMSBr (0.6 mmol), THF (0.5 mL), rt, 24 h. Isolated yields are given. [a]Yield of gram-scale reaction with **1a** (10 mmol, 1.36 g). [b]*Tert*-butylmagnesium bromide (1 mmol) was employed

reaction between benzophenone and *tert*-butyl Grignard reagent did not afford the alkylated product. It was observed that primary and secondary alkylmagnesium bromide cannot react with *N*-methylbenzamide to give the alkylated compounds. Interestingly, the formation of *para*-TMS-substituted benzamide in low yield was obversed when using isopropyl Grignard reagent in the alkylation.

### Discussion
To probe the possibility of forming low-valent chromium species, the reaction of CrCl₃ with *t*-BuMgBr was performed at room temperature (Fig. 4a). The evolution of hydrogen gas was

observed and quantified by GC/MS analyses. The amount of hydrogen evolution was nearly equal to the amount of chromium (III) salt, which indicates that the formation of a low-valent chromium species through transmetalation of CrCl₃ followed by β-Hydride elimination and hydride reduction could be considered. After the evolution of hydrogen, benzamide and TMSBr were added and the *para*-alkylation also occurred effectively to form **3** in 65% yield, suggesting that the reactive Cr species generated in situ could promote the transformation (Fig. 4b).

The reaction of *N,N'*-dimethylbenzamide (**44**) did not form the product, confirming that deprotonation of the NH group in

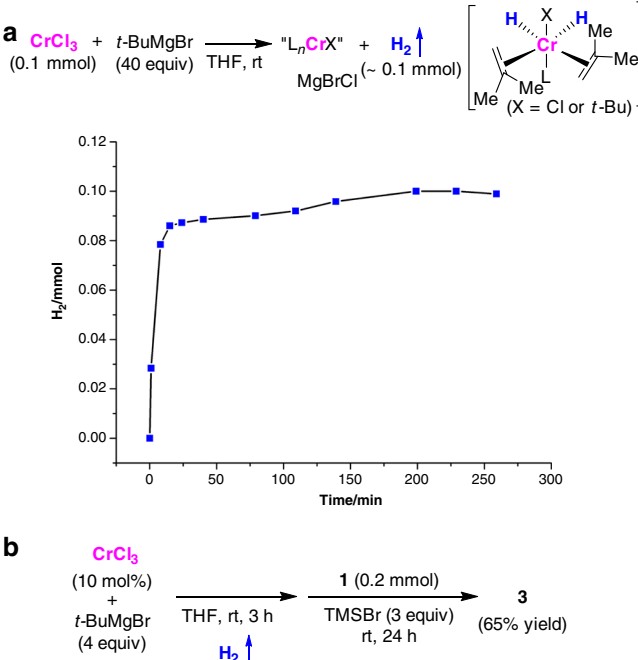

**Fig. 3** Cr-catalyzed *para*-alkylation of benzamides with tertiary alkylmagnesium bromides. Conditions: benzamide (0.2 mmol), tertiary alkylmagnesium bromide (0.8 mmol), CrCl$_3$ (0.02 mmol), TMSBr (0.6 mmol), THF (0.5 mL), rt, 24 h. Isolated yields are given

**Fig. 4** The formation of low-valent chromium species in situ for the *para*-selective alkylation. **a** Hydrogen evolution by the reaction of CrCl$_3$ with *t*-BuMgBr. **b** *para*-Selective alkylation of benzamide that was promoted by the in situ generated low-valent Cr species

benzamide by Grignard reagent to give benzimidate species is required to achieve the *para*-alkylation (Fig. 5a). The reaction of trimethylsilyl (*Z*)-*N*-phenylbenzimidate (**46**) with *t*-BuMgBr allowed the formation of the *para*-alkylated product **41** with Cr catalysis (Fig. 5b); whereas, the alkylation did not occur in the absence of either CrCl$_3$ or TMSBr (Fig. 5c, d). These result shows that Cr and TMSBr play important roles in the latter transformation of *para*-C–H bond. Like Nakao's reaction, the imino group on the benzimidate intermediate could ligate with the metal, and the coordination may enhance the reactivity of the electron-poor aromatic at the *para* position toward functionalization[21,41].

Quenching the alkylation by D$_2$O showed that almost no deuterium was incorporated into the *ortho*- or *para*-positions of the product **3** and starting material **1**. The related C–H bonds may not be metalated under present conditions (Fig. 5e). It was noteworthy that the addition of free radical inhibitor such as 2,2,6,6-tetramethyl-1-piperidinyloxy (TEMPO) into the reaction shut down the alkylation (Fig. 5f)[42–47]. The analysis of the alkylation of **1a** after 4 h using EPR spectroscopy suggests the formation of radical species during this reaction (see Supplementary Fig. 5 for details)[48–50]. Meanwhile, radical species in the reaction of *t*-BuMgBr with CrCl$_3$ and TMSBr was detected by EPR study[51]. Based on previous reports and these experimental results, the alkylation may proceed by a radical-type *para*-nucleophilic substitution of imino-coordination benzimidate to afford aryl radical, which undergoes a single electron transfer (SET)/proton transfer process to form *para*-alkylated compound (Fig. 5i)[52,53]. As to the role of trimethylsilyl bromide, in addition to the formation of benzimidate intermediate **46**, we hypothesized that it

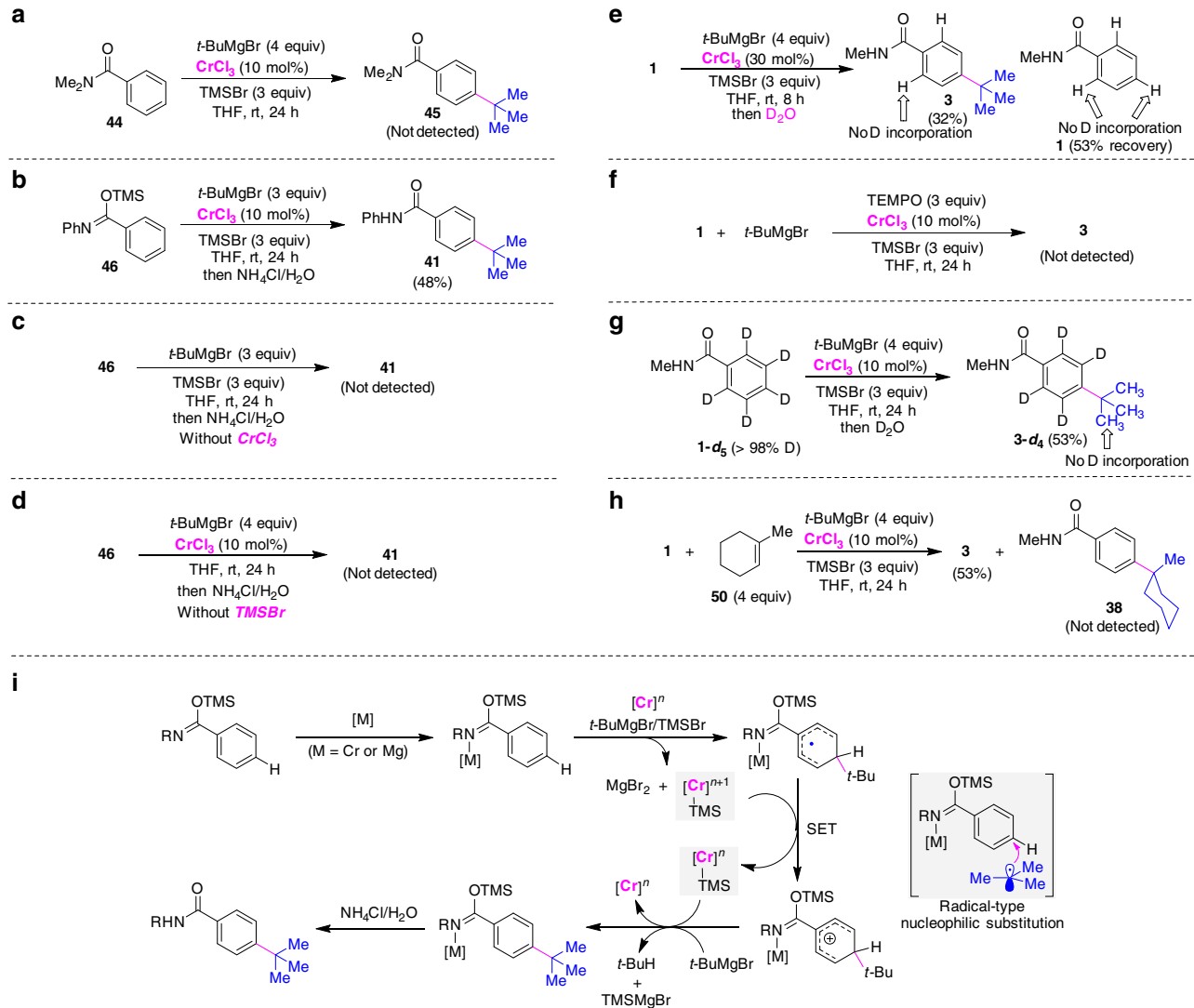

**Fig. 5** Preliminary mechanistic studies. **a** Alkylation with *N,N*-dimethyl-substituted benzamide. **b** Alkylation with *N*-phenylbenzimidate. **c** Alkylation without CrCl₃ salt. **d** Alkylation without TMSBr. **e** Quenching the alkylation using D₂O. **f** Scavenger experiment with TEMPO. **g** Alkylation with deuterated *N*-methylbenzamide. **h** Alkylation with 1-methylcyclohex-1-ene. **i** Plausible reaction pathways

may help to give *tert*-butyl radical and TMS-Cr intermediate by reaction with *t*-BuMgBr and low-valent Cr species in the catalytic cycle. The observation of *para*-TMS-substituted benzamide compound when using *i*-PrMgBr may indicate that the formation of TMS-Cr intermediate could be considered. It was found that no deuterium was incorporated into the *tert*-butyl group in **3-d₄**, and adding 1-methylcyclohex-1-ene into the reaction did not give hydroarylation products such as compound **38** (Fig. 5g, h). The addition of *para*-C–H bonds across olefin would not be involved in the alkylation. On the other hand, a small kinetic isotope effect (KIE ≈ 1.4) was observed in the Cr-catalyzed *para*-alkylation.

When kinetic studies on the reaction between **1a** with *t*-BuMgBr were carried out, the data revealed a positive first-order dependence of the alkylation on the concentration of CrCl₃ (Fig. 6a). This result suggests that the concentration of chromium likely determines the reaction rate and that a unimolecular event can be considered the turnover-liming step. The linear plot of log ($k_{obs}$) versus the logarithm of [**1**] showed the reaction to have a slight first-order dependence on the concentration of benzamide (Fig. 6b).

In conclusion, we have developed the *para*-selective alkylation of benzamide derivatives with chromium catalysis for

the formation of arylated quaternary carbon centers. The use of low-cost chromium(III) salt as precatalyst combined with trimethylsilyl bromide allowed the alkylative reaction to occur smoothly at room temperature. The methodology provides a selective way to incorporate bulky tertiary alkyl groups into the *para*-position of benzamide derivatives without either isomerization or *ortho*-alkylation. The presented catalytic activity of low-valent Cr function as a redox shuttle in the *para*-selective formation of quaternary carbon centers should spur the development of synthetic strategies with chromium.

## Methods

**Cr-catalyzed *para*-selective formation of quaternary centers**. A dried Schlenk tube was charged with *N*-methylbenzamide **1** (0.2 mmol), CrCl₃ (3 mg, 0.02 mmol) and freshly distilled THF (0.5 mL). Tertiary alkylmagnesium bromide **2** (0.8–1 mmol) was dropwise added by syringe at room temperature. After stirring the mixture for 30 min, trimethylbromosilane (92 mg, 0.6 mmol) was added by syringe and the mixture was stirred at room temperature for 24 h. The resulting mixture was then quenched by an aqueous solution of NH₄Cl and extraction with ethyl acetate (3 × 10 mL). The combined organic phase was dried over anhydrous Na₂SO₄ and concentrated under vacuum. The crude product was purified by silica gel chromatography to give the *para*-alkylated product **3**.

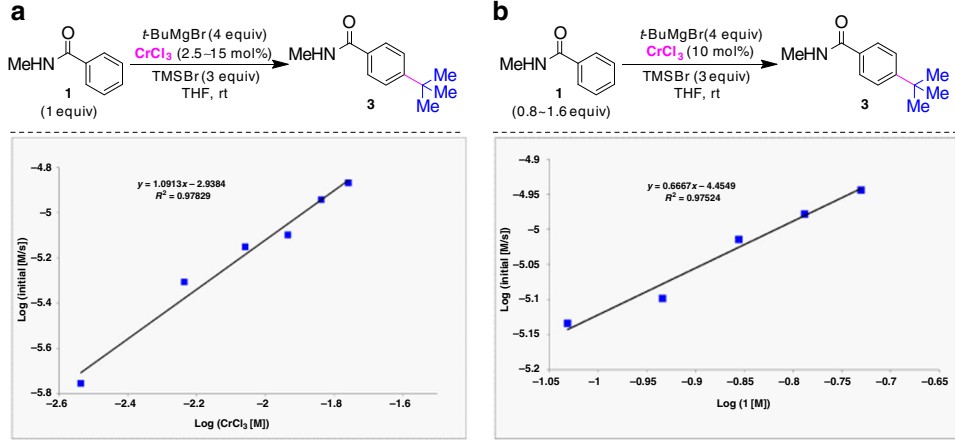

**Fig. 6** Kinetic profile for chromium-catalyzed *para*-alkylative reaction. **a** Plot of initial rate vs concentration of CrCl$_3$ reveal first-order kinetics for chromium salt. **b** Plot of initial rate vs concentration of **1** indicates a slightly first-order kinetics for benzamide

**Spectroscopic methods**. $^1$H and $^{13}$C NMR spectra were recorded on a Bruker DRX-400 (operating at 400 MHz for $^1$H and 100 MHz for $^{13}$C). UV–vis spectra were recorded on a Thermo Fisher Nicolet 6700 FT-IR spectrometer using ATR (Attenuated Total Reflectance) method.

**Single-crystal X-ray structure determinations**. The crystal data of **3a** were collected on a Bruker SMART CCD diffractometer with MoKα radiation (λ = 0.71073 Å). The structures were solved by direct methods and refined on $F^2$ using SHELXTL. All non-hydrogen atoms were refined anisotropically.

## Data availability
The X-ray crystallographic coordinates for structures that support the findings of this study have been deposited at the Cambridge Crystallographic Data Centre (CCDC) with the accession code CCDC 1821836 (**3**). The authors declare that all other data supporting the findings of this study are available within the article and Supplementary Information files, and also are available from the corresponding author upon reasonable request.

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

## Acknowledgements

The support for this work by the National Natural Science Foundation of China (Nos. 21202128 and 21572175), SCU from a start-up fund and Beijing National Laboratory for Molecular Sciences is gratefully acknowledged.

## Author contributions

P.L., C.C. and X.C. performed the experiments and analysed the data. J.T. and X.Z. wrote the manuscript. P.L., C.C. and X.C. contributed equally. All the authors discussed the results and commented on the manuscript.

## Additional information

**Competing interests:** The authors declare no competing interests.

