## [Peer Review File · Nature Communications]

Reviewer #1 (Remarks to the Author):

The manuscript by Zeng and coworkers demonstrates para-selective alkylation of benzamide by Cr catalysis through coupling with tert-alkyl Grignard reagents in the presence of TMSBr. The reaction represents an interesting and rare transformation allowing construction of quaternary carbon selectively at the para-position of benzamides. This could also be viewed as a very nice extension of the Minisci reaction, which is limited to heteroarenes, to substituted benzenes. The Cr catalyst system, which seems simple and thus practically easy to work with, is also unique and interesting for further studies. Overall, the manuscript would be interesting to broad audiences in the synthetic and organometallic communities, and the reviewer can support its publication in Nature Communications, provided that the following points are addressed.

1. What is DCB (entry 2 of Table 1) ?

2. What if primary amides, ketones, and carboxylic acids are reacted ?

3. What if other (1° and 2°) alkyl Grignard reagents are reacted ?

4. The transformation is oxidative coupling of arenes and Grignard reagents. It seems that TMSBr can be an only possible stoichiometric oxidant in this system. What is the fate of TMSBr ? The role of TMSBr except for the formation of intermediate 6 should be discussed and included in the plausible mechanism (Scheme 4).

5. Some of the compounds do not meet the standard for analytical purity (>95%). For example, 1H and 13C NMR spectra of 3l (silicon grease ?), 3m (3.3 ppm ?), 3n (1.25 ppm ?), and 3q (1.0 ppm ?) still contain certain amounts of unidentified impurities to affect the numbers given as "isolated yields".

Reviewer #2 (Remarks to the Author):

The paper describes a catalytic, para-selective formation of quaternary centres of benzamide derivatives. There are some attractive features of the presented methodology in that it employs a low-cost Chromium(III) salt, the reaction is efficient at room temperature and it is highly-selective for the para- position with no competitive meta/ortho products. The use of air-sensitive tertiary organometallic reagents is required which limits the scope of the methodology. Indeed compared with many state-of-the art catalytic C-H functionalisation processes the range of examples presented here is somewhat limited.

The methodology and experimental notes are thorough and detailed. Other researchers would easily be able to repeat the methods.

The authors devote considerable efforts to exploring the mechanism and carry out appropriate studies to establish a likely involvement of a radical species and a nucleophilic radical-type para-nucleophilic substitution. These are all routine experiments that are commonly carried out in this area. There is nothing surprising or unusual about the results.

This type of process is not particularly novel and there are clearly similar examples in the literature (a) J.-M. Li, Y.-H. Wang, Y. Yu, R.-B. Weng, G. Lu, *ACS Catal.*, 2017, 7, 2661. b) S. Liang, M. Bolte, G. Manolikakes, *Chem. Eur. J.*, 2017, 23, 96. c) Leitch, J. A.; Bhonoah, Y.; Frost, C. G. *ACS Catal.* 2017, 7, 5618.

Relevant examples of para-selective C-H functionalisation include: a) W. Liu, L. Ackermann, *Org. Lett.*, 2013, 15, 3484. b) X. Wang, D. Leow, J.-Q. Yu, *J. Am. Chem. Soc.*, 2011, 133, 13864. c) B. Ma, Z. Chu, B. Huang, Z. Liu, L. Liu, J. Zhang, *Angew. Chem. Int. Ed.*, 2017, 56, 2749. d) Y. X. Luan, T. Zhang, W.-W. Yao, K. Lu, L.-Y. Kong, Y.-T. Lin, M. Ye, *J. Am. Chem. Soc.*, 2017, 139, 1786. e) B. Berzina, I. Sokolovs, E. Suna, *ACS Catal.*, 2015, 5, 7008. f) L. T. Ball, G. C. Lloyd-Jones, C. A. Russell, *Science*, 2012, 337, 1644.

I do not think that there is anything presented in this paper that is original enough to influence a major change of thinking or direction in the field. It is significant that low-cost Chromium complexes can be employed and that is practically advantageous but not particularly original. I do not believe that this manuscript is suitable for *Nature Communications* - it would be more suitable for a specialist catalysis or synthetic methodology journal.

Reviewer #3 (Remarks to the Author):

In the submitted manuscript, Zeng et al report a Cr-catalyzed C-H functionalization process that incorporates tertiary alkyl groups selectively into the para position of benzamide derivatives. This work builds upon previous work from the same research group on the use of low-valent chromium species as catalysts in organic synthesis. The formation of aryl-substituted quaternary centers via metal-catalyzed processes is very challenging, and typically occurs with concurrent formation of isomerized products via beta-hydride elimination/reinsertion pathways. These isomerized products often form in significant quantities, and are difficult/impossible to separate from the desired product. The process described in this manuscript enables the formation of such quaternary centers without the formation of isomerized side products. Additionally, in place of aryl halide electrophiles, a C-H functionalization strategy is employed, which is completely selective for the para position of benzamide substrates. While yields do not tend to be particularly high, the selectivity exhibited in this reaction is impressive. This enables access to hindered products that are inaccessible using existing coupling methods. The formation of para-selective t-butylated products even with a meta substituent is present (e.g., 3e and 3f) demonstrates the power of this method. I support this manuscript for publication in Nature Communications after some considerations are addressed. My major issue with the manuscript in its current form relates to its mechanistic component (see below for more details).

The formation of a catalytically competent species by pretreating the chromium salt with t-BuMgBr is advanced as evidence that the chromium salt is directly activated via reaction with t-BuMgBr (Figure 3). However, does the subsequent addition of TMSBr affect the formation of the actual catalytic species? How does TMSBr participate in other steps of the catalytic process? In equation 4 of Figure 4a, the authors also show that TMSBr is required in the catalytic process. This really requires elaboration. On page S30 of the SI, the authors indicate that an analogous reaction using iPrMgBr failed to generate the alkylation product, but did generate 20% of a para-silylation product. This is a very interesting observation that may have important mechanistic implications. From the combined data generated by the mechanistic probes, a mechanism is proposed in Figure 4b. I am very confused by elements of this mechanism. Wouldn't an initial oxidative process be required to generate a tBu radical from tBu-MgBr? This should result in a reduced Cr species. However, an oxidized species (Cr(n+1)) is indicated in Figure 4. This reaction should amount to a 2-electron oxidation process (t-butyl anion is formally substituting an H+). From the proposed mechanism, I don't see how this is occurring. Again, TMSBr may be involved in this part of the reaction, but this isn't shown in the mechanism. Otherwise, why would low valent Cr be required in a mechanism in which oxidation of t-BuMgBr is occurring?

Other comments:

Is the formation of olefins observed when heavier Grignard reagents are used? This would also provide support for a beta-hydride elimination activation pathway.

Are saturation kinetics observed when excess benzamide is employed? This could indicate a pre-equilibrium with benzamide coordinating to the active chromium catalyst.

In the description of conditions used in Figure 2, I think that "TMSBr (0.06 mmol)" should be "TMSBr (0.6 mmol)."

"Interestingly, the para-alkylation with 10 mol % of FeCl₃, AlCl₃ or AlMe₃ also occurred, albeit with giving low conversions (entries 11 and 12)" should probably be "Interestingly, the para-alkylation with 10 mol % of FeCl₃, AlCl₃ or AlMe₃ also occurred, albeit with low conversions (entries 11 and 12)."

On page 10, the authors write: "The analysis of the alkylation of 1a after 4 hours by EPR spectroscopy technique showed that radical intermediates are involved in the reaction." EPR may show the presence of radical species, but this does not prove that they are actually involved in the active catalytic cycle. I would re-phrase this sentence: "The analysis of the alkylation of 1a after 4 hours using EPR spectroscopy suggests the formation of radical species during this reaction."

Page 10: "singlet electron transfer" should be "single electron transfer."

Are silylated products ever observed when tBuMgBr is used? The NMRs for several products (e.g., S85) suggest the possible presence of a silylated byproduct.

Response to the Reviewer 1's comments

The manuscript by Zeng and coworkers demonstrates para-selective alkylation of benzamide by Cr catalysis through coupling with tert-alkyl Grignard reagents in the presence of TMSBr. The reaction represents an interesting and rare transformation allowing construction of quaternary carbon selectively at the para-position of benzamides. This could also be viewed as a very nice extension of the Minisci reaction, which is limited to heteroarenes, to substituted benzenes. The Cr catalyst system, which seems simple and thus practically easy to work with, is also unique and interesting for further studies.

We sincerely thank this Reviewer for the positive evaluation and comments on this work.

Overall, the manuscript would be interesting to broad audiences in the synthetic and organometallic communities, and the reviewer can support its publication in Nature Communications, provided that the following points are addressed.

We thank the Reviewer for the strongly supporting for publication.

1. What is DCB (entry 2 of Table 1) ?

DCB is the compound of 2,3-dichlorobutane. Now it was clearly presented in the text of the manuscript by "2,3-dichlorobutane (2,3-DCB)". Meanwhile, the name of this compound was shortening to be "2,3-DCB" in entry 2 of Table 1.

2. What if primary amides, ketones, and carboxylic acids are reacted ?

As showing in the following Scheme, the Cr-catalyzed alkylation using primary amide (**1al**), aromatic ketone (**1am**), benzoic acid (**1an**) and *N*-methyl-1-naphthamide (**1ao**) did not form the desired para-alkylated products. In most cases, the starting materials can be recovered. In the reaction with benzophenone (**1am**), the addition of Grignard reagent into carbonyl occurred to give the related alcohol product. In addition, the use of *N*-methyl-1*H*-pyrrole-2-carboxamide (**1ap**) and *N*-methylfuran-2-carboxamide (**1aq**) in the alkylation did not give the alkylated products.

Scheme S1. Inefficient Substrates in the Cr-Catalyzed *para*-Alkylation with *t*-BuMgBr

Inefficient substrates in the Cr-catalyzed *para*-alkylation by reaction with *t*-BuMgBr

Inefficient substrates in the Cr-catalyzed alkylation by reaction with *t*-BuMgBr

3. What if other (1° and 2°) alkyl Grignard reagents are reacted ?

As shown in the following equations, the use of 1° alkyl Grignard reagent of MeMgBr did not give the *para*-alkylated product under Cr catalysis. Interestingly, the Cr-catalyzed alkylation with 2° alkyl Grignard reagent of *t*-PrMgBr formed the *para*-silylated compound in 20% yield. Now we tried to improve the transformation by screening the reaction parameters.

4. The transformation is oxidative coupling of arenes and Grignard reagents. It seems that TMSBr can be an only possible stoichiometric oxidant in this system. What is the fate of TMSBr ? The role of TMSBr except for the formation of intermediate 6 should be discussed and included in the plausible mechanism (Scheme 4).

Thank the Reviewer for the valuable comments.

As shown in the following Scheme, in addition to the formation of benzimidate intermediate, TMSBr may help to the formation of *tert*-butyl radical by the reaction with *t*-BuMgBr and low-valent Cr, leading to (TMS)CrL_{*n*} intermediate and MgBr₂ salt. Notably, the reaction with *t*-PrMgBr was able to give *para*-silylated product in low yield. This indicates that the formation of (TMS)Cr species or silyl radical can be considered. After undergoing a single electron transfer process, the formation of TMSMgBr might be considered by reaction with *t*-BuMgBr and aryl cation (please see Scheme S2 for detail).

The comment of “As to the role of trimethylsilyl bromide, in addition to the formation of benzimidate intermediate 46, we hypothesized that it may help to give *tert*-butyl radical by reaction with *t*-BuMgBr and low-valent Cr species in the catalytic cycle” was supplemented into the revised manuscript.

Scheme S2. Proposed Mechanism

5. Some of the compounds do not meet the standard for analytical purity (>95%). For example, ¹H and ¹³C NMR spectra of **3l** (silicon grease ?), **3m** (3.3 ppm ?), **3n** (1.25 ppm ?), and **3q** (1.0 ppm ?) still contain certain amounts of unidentified impurities to affect the numbers given as "isolated yields".

Thank the Reviewer for pointing out the NMR spectra problem.

We repeated the related *para*-alkylation. The relevant crude products were purified again by silica gel chromatography. Although great efforts have been directed to remove the silicon grease for compound **3l** (named in previous version), the related ¹H and ¹³C NMR spectra still show the signals of silicon grease impurities (please see the following NMR spectra).

We found that the impurities for the mentioned signals by this Reviewer showing in the spectra of compounds **3m** (3.3 ppm) and **3q** (1.0 ppm) were removed.

Although considerable efforts have been directed in the purification of compound **3n**, the ¹H NMR spectrum in 1.25 ppm still shows little bit unidentified impurities, but the related ¹³C NMR spectra is very clean.

Figure S1. ¹H and ¹³C NMR spectra for compound **3l**.

Figure S2. ¹H NMR spectra for compound **3m**.

Figure S3. ¹H NMR spectra for compound **3q**.

Figure S4. ¹H and ¹³C NMR spectra for compound **3n**.

Response to the Reviewer 2's comments

The paper describes a catalytic, para-selective formation of quaternary centres of benzamide derivatives. There are some attractive features of the presented methodology in that it employs a low-cost Chromium(III) salt, the reaction is efficient at room temperature and it is highly-selective for the para- position with no competitive meta/ortho products.

We thank the Reviewer for the positive evaluation on this work.

The use of air-sensitive tertiary organometallic reagents is required which limits the scope of the methodology. Indeed compared with many state-of-the art catalytic C-H functionalisation processes the range of examples presented here is somewhat limited. The methodology and experimental notes are thorough and detailed. Other researchers would easily be able to repeat the methods.

Thank the Reviewer for the comments.

The authors devote considerable efforts to exploring the mechanism and carry out appropriate studies to establish a likely involvement of a radical species and a nucleophilic radical-type para-nucleophilic substitution. These are all routine experiments that are commonly carried out in this area. There is nothing surprising or unusual about the results.

This type of process is not particularly novel and there are clearly similar examples in the literature (a) J.-M. Li, Y.-H. Wang, Y. Yu, R.-B. Weng, G. Lu, ACS Catal., 2017, 7, 2661. b) S. Liang, M. Bolte, G. Manolikakes, Chem. Eur. J., 2017, 23, 96. c) Leitch, J. A.; Bhonoah, Y.; Frost, C. G. ACS Catal. 2017, 7, 5618.

Relevant examples of para-selective C-H functionalisation include: a) W. Liu, L. Ackermann, Org. Lett., 2013, 15, 3484. b) X. Wang, D. Leow, J.-Q. Yu, J. Am. Chem. Soc., 2011, 133, 13864. c) B. Ma, Z. Chu, B. Huang, Z. Liu, L. Liu, J. Zhang, Angew. Chem. Int. Ed., 2017, 56, 2749. d) Y. X. Luan, T. Zhang, W.-W. Yao, K. Lu, L.-Y. Kong, Y.-T. Lin, M. Ye, J. Am. Chem. Soc., 2017, 139, 1786. e) B. Berzina, I. Sokolovs, E. Suna, ACS Catal., 2015, 5, 7008. f) L. T. Ball, G. C. Lloyd-Jones, C. A. Russell, Science, 2012, 337, 1644.

Thank the reviewer for the mention of these references.

As the comments of the Reviewer, these references are related to *para*-C–H bond functionalization. Through carefully checking these references, yet we found that they have no relationship with synthetic methods of the construction of quaternary carbon centers. We are afraid that this Reviewer may not fully catch the impressive points of this work description in the manuscript. ***In addition to the amazed para-selectivity, as the comment of Reviewer 2, the described method here can be used to the formation of arylated quaternary carbon centers without the formation of isomerized side products, which remains a formidable and long-standing challenge in synthetic chemistry because of the competing beta-hydride elimination/reinsertion pathways under transition metal catalysis.*** This is a prominent advantage of this method when compared with previous examples for the construction quaternary carbon centers with palladium catalysis. In particular, this reaction enables access to hindered products that are inaccessible using existing coupling methods. The formation of para-selective *t*-butylated products even with a meta substituent is present demonstrates the power of this method.

I do not think that there is anything presented in this paper that is original enough to influence a major change of thinking or direction in the field. It is significant that low-cost Chromium complexes can be employed and that is practically advantageous but not particularly original.

We thank the Reviewer for mention that “*It is significant that low-cost chromium complexes can be employed and that is practically advantageous*”.

I do not believe that this manuscript is suitable for Nature Communications - it would be more suitable for a specialist catalysis or synthetic methodology journal.

Thank the Reviewer for the comments.

Response to the Reviewer 3's comments

In the submitted manuscript, Zeng et al report a Cr-catalyzed C-H functionalization process that incorporates tertiary alkyl groups selectively into the para position of benzamide derivatives. This work builds upon previous work from the same research group on the use of low-valent chromium species as catalysts in organic synthesis. The formation of aryl-substituted quaternary centers via metal-catalyzed processes is very challenging, and typically occurs with concurrent formation of isomerized products via beta-hydride elimination/reinsertion pathways. These isomerized products often form in significant quantities, and are difficult/impossible to separate from the desired product. The process described in this manuscript enables the formation of such quaternary centers without the formation of isomerized side products.

We sincerely thank the Reviewer for the positive evaluation on this work.

Additionally, in place of aryl halide electrophiles, a C-H functionalization strategy is employed, which is completely selective for the para position of benzamide substrates. While yields do not tend to be particularly high, the selectivity exhibited in this reaction is impressive. This enables access to hindered products that are inaccessible using existing coupling methods. The formation of para-selective *t*-butylated products even with a meta substituent is present (e.g., 3e and 3f) demonstrates the power of this method. I support this manuscript for publication in Nature Communications after some considerations are addressed.

We thank the Reviewer for the strongly supporting.

My major issue with the manuscript in its current form relates to its mechanistic component (see below for more details).

The formation of a catalytically competent species by pretreating the chromium salt with *t*-BuMgBr is advanced as evidence that the chromium salt is directly activated via reaction with *t*-BuMgBr (Figure 3). However, does the subsequent addition of TMSBr affect the formation of the actual catalytic species? How does TMSBr participate in other steps of the catalytic process? In equation 4 of Figure 4a, the authors also show that TMSBr is required in the catalytic process. This really requires elaboration.

Thank the Reviewer for the valuable comment and question.

In view of that TMSBr is required for the *para*-alkylation of benzimidate with chromium catalysis, we hypothesized that TMSBr might react with *t*-BuMgBr and low-valent Cr to form *tert*-butyl radical combined with (TMS)Cr species and MgBr₂ salt, making the radical-type nucleophilic substitution occurring at the para position of benzimidate.

The resulting aryl radical may undergo a single electron transfer (SET) process by loss of a single electron to form aryl cation, in which (TMS)Cr species can serve as electron acceptor followed by the reaction with *t*-BuMgBr and aryl cation to give *t*-BuH, TMSMgBr and regeneration of low-valent L_nCr species.

On page S30 of the SI, the authors indicate that an analogous reaction using $iPrMgBr$ failed to generate the alkylation product, but did generate 20% of a *para*-silylation product. This is a very interesting observation that may have important mechanistic implications.

In the reaction with $tPrMgBr$, the *para*-silylated product was formed rather than the related alkylation product. This result indicates that the formation of $(TMS)Cr$ organometallic species or silyl radical can be considered followed by *para*-reaction with benzimidate to give the related silylation product. On the other hand, because of the relatively instability of isopropyl radical as compared with the *tert*-butyl one, it might be unfavorable to form isopropyl radical, resulting in the inefficiency when using $tPrMgBr$ in the reaction.

From the combined data generated by the mechanistic probes, a mechanism is proposed in Figure 4b. I am very confused by elements of this mechanism. Wouldn't an initial oxidative process be required to generate a *t*-Bu radical from $t-BuMgBr$?

Yes, an initial oxidative process is required to generate a *tert*-butyl radical from $t-BuMgBr$.

This should result in a reduced Cr species. However, an oxidized species ($Cr(n+1)$) is indicated in Figure 4. This reaction should amount to a 2-electron oxidation process (*t*-butyl anion is formally substituting an H^+).

We agree that this should result in the formation of a reduced species in the process. In the presence of low-valent $Cr(n)$, we envisioned that $TMSBr$ might serve as electron acceptor to give bromide anion (the formation of $MgBr_2$ salt) and silyl radical, which combines with low-valent $Cr(n)$ to form $(TMS)Cr(n+1)$ species. It may explain why $TMSBr$ is required for achieving the latter *para*-alkylation with benzimidate intermediate. After the process of single electron transfer, *t*-butyl anion of Grignard reagent should abstract the proton by reaction with aryl cation and $(TMS)Cr$, leading to $TMSMgBr$ and regeneration of low-valent Cr .

From the proposed mechanism, I don't see how this is occurring. Again, $TMSBr$ may be involved in this part of the reaction, but this isn't shown in the mechanism. Otherwise, why would low valent Cr be required in a mechanism in which oxidation of $t-BuMgBr$ is occurring?

We thank the Reviewer for the valuable comments and questions. $TMSBr$ should play important role and be involved in this part of the reaction. We revised the related elementary step and shows the formation of $(TMS)Cr(n+1)$ species in the revised manuscript. It may provide an explanation why low-valent Cr species is required in a mechanism in which oxidation of $t-BuMgBr$ is occurring.

Other comments:

Is the formation of olefins observed when heavier Grignard reagents are used? This would also provide support for a beta-hydride elimination activation pathway.

Thank the Reviewer for the question. Indeed, we observed that small amount of related olefin was formed by GC-MS analysis when using Grignard reagent of (1-methylcyclohexyl)magnesium bromide in the alkylation (in case of forming product **38**).

Are saturation kinetics observed when excess benzamide is employed? This could indicate a pre-equilibrium with benzamide coordinating to the active chromium catalyst.

We thank the Reviewer for the valuable question. We studied the dependence of the initial reaction rate on *N*-methylbenzamide (**1**) with high concentrations. The reaction shows first-order kinetics for benzamide when the concentration was below ~ 0.19 M. It seems that the dependence of the reaction rate on the concentration of **1** was more complex with a high concentration (Please see the following Table for the related data). The reaction rate was significantly decreased when the concentration of **1** was higher than ~ 0.19 M. However, the alkylation almost cannot occur when increasing the concentration of **1** was up to 0.35 M. It's hard to say that the initial rate of benzamide exhibited a saturation behavior at high concentrations.

Table S5. Initial Rate Data Obtained by Varying the Concentration of 1

Entry	amide 1 [M]	^t BuMgBr 2 [M]	Initial rate [M/min]
1	0.0930	0.4761	4.41×10^{-4}
2	0.1163	0.4761	4.78×10^{-4}
3	0.1395	0.4761	5.81×10^{-4}
4	0.1628	0.4761	6.31×10^{-4}
5	0.1860	0.4761	6.83×10^{-4}
6	0.2326	0.4761	3.57×10^{-4}
7	0.3488	0.4761	0

In the description of conditions used in Figure 2, I think that “TMSBr (0.06 mmol)” should be “TMSBr (0.6 mmol).”

We thank the Reviewer for pointing out this error. It was corrected in the revised Figures 2 and 3.

“Interestingly, the para-alkylation with 10 mol % of FeCl₃, AlCl₃ or AlMe₃ also occurred, albeit with giving low conversions (entries 11 and 12)” should probably be “Interestingly, the para-alkylation with 10 mol % of FeCl₃, AlCl₃ or AlMe₃ also occurred, albeit with low conversions (entries 11 and 12).”

It has been corrected in the revised manuscript.

On page 10, the authors write: “The analysis of the alkylation of **1a** after 4 hours by EPR spectroscopy technique showed that radical intermediates are involved in the reaction.” EPR may show the presence of radical species, but this does not prove that they are actually involved in the

active catalytic cycle. I would re-phrase this sentence: “The analysis of the alkylation of 1a after 4 hours using EPR spectroscopy suggests the formation of radical species during this reaction.”

Thank the Reviewer for the kindly suggestion. This sentence was re-written accordingly.

Page 10: “singlet electron transfer” should be “single electron transfer.”

We thank the Reviewer for pointing out this mistake. It was corrected in the revised manuscript.

Are silylated products ever observed when *t*BuMgBr is used? The NMRs for several products (e.g., S85) suggest the possible presence of a silylated byproduct.

Thank the Reviewer for the questions. The silylated products were not observed when the use of *t*-BuMgBr in the reaction. The appeared signals around 0 ppm in NMR spectra of several products may attribute to silicon grease. Other signals in the ¹H and ¹³C NMR spectra are consistent with the desired products.

Reviewer #1 (Remarks to the Author):

The revised manuscript now tries to address some of the issues raised by the reviewers. In particular, it was very important to address the role of TMSBr. In the revised manuscript, it is proposed to play a role in the formation of tBu radical. It is shown that TMS-[Cr] species is formed in the mechanistic scheme. This proposal may be supported by the observation that para-TMS-substituted benzamide is formed when iPr-MgBr is reacted. This result should be included in the main text because it can partially support the proposed mechanism. Also, the results with 1° and 2° alkyl Grignard reagents as well as other aromatic carbonyl compounds should also be mentioned in the main text. The authors should note that questions and comments from the reviewers should be regarded as representative of those that might be posed by readers of the article. Consequently, these questions and comments should be addressed not only in the response letter but also within the body of the resubmitted manuscript itself.

With these revisions to improve the discussion about the mechanism in hand, the manuscript may warrant publication in Nature Communications.

Reviewer #3 (Remarks to the Author):

The revised manuscript has addressed the mechanistic questions that I originally had. The proposed mechanism is now more clearly depicted, and follow up mechanistic experiments have been conducted. I feel that this manuscript is now suitable for publication in Nature Communications.

One typo: In Figure 6, the y axes should be labeled "(initial)" not "(initiate)"

The First Cycle of Response to the Reviewer's Comments

Response to the Reviewer 1's comments

The manuscript by Zeng and coworkers demonstrates para-selective alkylation of benzamide by Cr catalysis through coupling with tert-alkyl Grignard reagents in the presence of TMSBr. The reaction represents an interesting and rare transformation allowing construction of quaternary carbon selectively at the para-position of benzamides. This could also be viewed as a very nice extension of the Minisci reaction, which is limited to heteroarenes, to substituted benzenes. The Cr catalyst system, which seems simple and thus practically easy to work with, is also unique and interesting for further studies.

We sincerely thank this Reviewer for the positive evaluation and comments on this work.

Overall, the manuscript would be interesting to broad audiences in the synthetic and organometallic communities, and the reviewer can support its publication in Nature Communications, provided that the following points are addressed.

We thank the Reviewer for the strongly supporting for publication.

1. What is DCB (entry 2 of Table 1) ?

DCB is the compound of 2,3-dichlorobutane. Now it was clearly presented in the text of the manuscript by "2,3-dichlorobutane (2,3-DCB)". Meanwhile, the name of this compound was shortening to be "2,3-DCB" in entry 2 of Table 1.

2. What if primary amides, ketones, and carboxylic acids are reacted ?

As showing in the following Scheme, the Cr-catalyzed alkylation using primary amide (**1al**), aromatic ketone (**1am**), benzoic acid (**1an**) and *N*-methyl-1-naphthamide (**1ao**) did not form the desired para-alkylated products. In most cases, the starting materials can be recovered. In the reaction with benzophenone (**1am**), the addition of Grignard reagent into carbonyl occurred to give the related alcohol product. In addition, the use of *N*-methyl-1*H*-pyrrole-2-carboxamide (**1ap**) and *N*-methylfuran-2-carboxamide (**1aq**) in the alkylation did not give the alkylated products.

Scheme S1. Inefficient Substrates in the Cr-Catalyzed *para*-Alkylation with *t*-BuMgBr

Inefficient substrates in the Cr-catalyzed *para*-alkylation by reaction with *t*-BuMgBr

Inefficient substrates in the Cr-catalyzed alkylation by reaction with *t*-BuMgBr

3. What if other (1° and 2°) alkyl Grignard reagents are reacted ?

As shown in the following equations, the use of 1° alkyl Grignard reagent of MeMgBr did not give the *para*-alkylated product under Cr catalysis. Interestingly, the Cr-catalyzed alkylation with 2° alkyl Grignard reagent of *i*-PrMgBr formed the *para*-silylated compound in 20% yield. Now we tried to improve the transformation by screening the reaction parameters.

4. The transformation is oxidative coupling of arenes and Grignard reagents. It seems that TMSBr can be an only possible stoichiometric oxidant in this system. What is the fate of TMSBr ? The role of TMSBr except for the formation of intermediate 6 should be discussed and included in the plausible mechanism (Scheme 4).

Thank the Reviewer for the valuable comments.

As shown in the following Scheme, in addition to the formation of benzimidate intermediate, TMSBr may help to the formation of *tert*-butyl radical by the reaction with *t*-BuMgBr and low-valent Cr, leading to (TMS)CrL_n intermediate and MgBr₂ salt. Notably, the reaction with *i*-PrMgBr was able to give *para*-silylated product in low yield. This indicates that the formation of (TMS)Cr species or silyl radical can be considered. After undergoing a single electron transfer process, the formation of TMSMgBr might be considered by reaction with *t*-BuMgBr and aryl cation (please see Scheme S2 for detail).

The comment of “As to the role of trimethylsilyl bromide, in addition to the formation of benzimidate intermediate **46**, we hypothesized that it may help to give *tert*-butyl radical by reaction with *t*-BuMgBr and low-valent Cr species in the catalytic cycle” was supplemented into the revised manuscript.

Scheme S2. Proposed Mechanism

5. Some of the compounds do not meet the standard for analytical purity (>95%). For example, ¹H and ¹³C NMR spectra of **3l** (silicon grease ?), **3m** (3.3 ppm ?), **3n** (1.25 ppm ?), and **3q** (1.0 ppm ?) still contain certain amounts of unidentified impurities to affect the numbers given as "isolated yields".

Thank the Reviewer for pointing out the NMR spectra problem.

We repeated the related *para*-alkylation. The relevant crude products were purified again by silica gel chromatography. Although great efforts have been directed to remove the silicon grease for compound **3l** (named in previous version), the related ¹H and ¹³C NMR spectra still show the signals of silicon grease impurities (please see the following NMR spectra).

We found that the impurities for the mentioned signals by this Reviewer showing in the spectra of compounds **3m** (3.3 ppm) and **3q** (1.0 ppm) were removed.

Although considerable efforts have been directed in the purification of compound **3n**, the ¹H NMR spectrum in 1.25 ppm still shows little bit unidentified impurities, but the related ¹³C NMR spectra is very clean.

Figure S1. ^1H and ^{13}C NMR spectra for compound **3l**.

Figure S2. ^1H NMR spectra for compound **3m**.

Figure S3. ¹H NMR spectra for compound **3q**.

Figure S4. ^1H and ^{13}C NMR spectra for compound **3n**.

Response to the Reviewer 2's comments

The paper describes a catalytic, para-selective formation of quaternary centres of benzamide derivatives. There are some attractive features of the presented methodology in that it employs a low-cost Chromium(III) salt, the reaction is efficient at room temperature and it is highly-selective for the para- position with no competitive meta/ortho products.

We thank the Reviewer for the positive evaluation on this work.

The use of air-sensitive tertiary organometallic reagents is required which limits the scope of the methodology. Indeed compared with many state-of-the art catalytic C-H functionalisation processes the range of examples presented here is somewhat limited. The methodology and experimental notes are thorough and detailed. Other researchers would easily be able to repeat the methods.

Thank the Reviewer for the comments.

The authors devote considerable efforts to exploring the mechanism and carry out appropriate studies to establish a likely involvement of a radical species and a nucleophilic radical-type para-nucleophilic substitution. These are all routine experiments that are commonly carried out in this area. There is nothing surprising or unusual about the results.

This type of process is not particularly novel and there are clearly similar examples in the literature (a) J.-M. Li, Y.-H. Wang, Y. Yu, R.-B. Weng, G. Lu, ACS Catal., 2017, 7, 2661. b) S. Liang, M. Bolte, G. Manolikakes, Chem. Eur. J., 2017, 23, 96. c) Leitch, J. A.; Bhonoah, Y.; Frost, C. G. ACS Catal. 2017, 7, 5618.

Relevant examples of para-selective C-H functionalisation include: a) W. Liu, L. Ackermann, Org. Lett., 2013, 15, 3484. b) X. Wang, D. Leow, J.-Q. Yu, J. Am. Chem. Soc., 2011, 133, 13864. c) B. Ma, Z. Chu, B. Huang, Z. Liu, L. Liu, J. Zhang, Angew. Chem. Int. Ed., 2017, 56, 2749. d) Y. X. Luan, T. Zhang, W.-W. Yao, K. Lu, L.-Y. Kong, Y.-T. Lin, M. Ye, J. Am. Chem. Soc., 2017, 139, 1786. e) B. Berzina, I. Sokolovs, E. Suna, ACS Catal., 2015, 5, 7008. f) L. T. Ball, G. C. Lloyd-Jones, C. A. Russell, Science, 2012, 337, 1644.

Thank the reviewer for the mention of these references.

As the comments of the Reviewer, these references are related to *para*-C-H bond functionalization. Through carefully checking these references, yet we found that they have no relationship with synthetic methods of the construction of quaternary carbon centers.

We are afraid that this Reviewer may not fully catch the impressive points of this work description in the manuscript. ***In addition to the amazed para-selectivity, as the comment of Reviewer 2, the described method here can be used to the formation of arylated quaternary carbon centers without the formation of isomerized side products, which remains a formidable and long-standing challenge in synthetic chemistry because of the competing beta-hydride elimination/reinsertion pathways under transition metal catalysis.*** This is a prominent advantage of this method when compared with previous examples for the construction quaternary carbon centers with palladium catalysis. In particular, this reaction enables access to hindered products that are inaccessible using existing coupling methods. The formation of para-selective *t*-butylated products even with a meta substituent is present demonstrates the power of this method.

I do not think that there is anything presented in this paper that is original enough to influence a major change of thinking or direction in the field. It is significant that low-cost Chromium complexes can be employed and that is practically advantageous but not particularly original.

We thank the Reviewer for mention that “*It is significant that low-cost chromium complexes can be employed and that is practically advantageous*”.

I do not believe that this manuscript is suitable for Nature Communications - it would be more suitable for a specialist catalysis or synthetic methodology journal.

Thank the Reviewer for the comments.

Response to the Reviewer 3's comments

In the submitted manuscript, Zeng et al report a Cr-catalyzed C-H functionalization process that incorporates tertiary alkyl groups selectively into the para position of benzamide derivatives. This work builds upon previous work from the same research group on the use of low-valent chromium species as catalysts in organic synthesis. The formation of aryl-substituted quaternary centers via metal-catalyzed processes is very challenging, and typically occurs with concurrent formation of isomerized products via beta-hydride elimination/reinsertion pathways. These isomerized products often form in significant quantities, and are difficult/impossible to separate from the desired product. The process described in this manuscript enables the formation of such quaternary centers without the formation of isomerized side products.

We sincerely thank the Reviewer for the positive evaluation on this work.

Additionally, in place of aryl halide electrophiles, a C-H functionalization strategy is employed, which is completely selective for the para position of benzamide substrates. While yields do not tend to be particularly high, the selectivity exhibited in this reaction is impressive. This enables access to hindered products that are inaccessible using existing coupling methods. The formation of para-selective *t*-butylated products even with a meta substituent is present (e.g., 3e and 3f) demonstrates the power of this method. I support this manuscript for publication in Nature Communications after some considerations are addressed.

We thank the Reviewer for the strongly supporting.

My major issue with the manuscript in its current form relates to its mechanistic component (see below for more details).

The formation of a catalytically competent species by pretreating the chromium salt with *t*-BuMgBr is advanced as evidence that the chromium salt is directly activated via reaction with *t*-BuMgBr (Figure 3). However, does the subsequent addition of TMSBr affect the formation of the actual catalytic species? How does TMSBr participate in other steps of the catalytic process? In equation 4 of Figure 4a, the authors also show that TMSBr is required in the catalytic process. This really requires elaboration.

Thank the Reviewer for the valuable comment and question.

In view of that TMSBr is required for the *para*-alkylation of benzimidate with chromium catalysis, we hypothesized that TMSBr might react with *t*-BuMgBr and low-valent Cr to form *tert*-butyl radical combined with (TMS)Cr species and MgBr₂ salt, making the radical-type nucleophilic substitution occurring at the para position of benzimidate.

The resulting aryl radical may undergo a single electron transfer (SET) process by loss of a single electron to form aryl cation, in which (TMS)Cr species can serve as electron acceptor followed by the reaction with *t*-BuMgBr and aryl cation to give *t*-BuH, TMSMgBr and regeneration of low-valent L_{*n*}Cr species.

On page S30 of the SI, the authors indicate that an analogous reaction using *i*PrMgBr failed to generate the alkylation product, but did generate 20% of a *para*-silylation product. This is a very interesting observation that may have important mechanistic implications.

In the reaction with *i*PrMgBr, the *para*-silylated product was formed rather than the related alkylation product. This result indicates that the formation of (TMS)Cr organometallic species or silyl radical can be considered followed by *para*-reaction with benzimidate to give the related silylation product. On the other hand, because of the relatively instability of isopropyl radical as compared with the *tert*-butyl one, it might be unfavorable to form isopropyl radical, resulting in the inefficiency when using *i*PrMgBr in the reaction.

From the combined data generated by the mechanistic probes, a mechanism is proposed in Figure 4b. I am very confused by elements of this mechanism. Wouldn't an initial oxidative process be required to generate a *t*-Bu radical from *t*-BuMgBr?

Yes, an initial oxidative process is required to generate a *tert*-butyl radical from *t*-BuMgBr.

This should result in a reduced Cr species. However, an oxidized species (Cr(n+1)) is indicated in Figure 4. This reaction should amount to a 2-electron oxidation process (*t*-butyl anion is formally substituting an H+).

We agree that this should result in the formation of a reduced species in the process. In the presence of low-valent Cr(n), we envisioned that TMSBr might serve as electron acceptor to give bromide anion (the formation of MgBr₂ salt) and silyl radical, which combines with low-valent Cr(n) to form (TMS)Cr(n+1) species. It may explain why TMSBr is required for achieving the latter *para*-alkylation with benzimidate intermediate. After the process of single electron transfer, *t*-butyl anion of Grignard reagent should abstract the proton by reaction with aryl cation and (TMS)Cr, leading to TMSMgBr and regeneration of low-valent Cr.

From the proposed mechanism, I don't see how this is occurring. Again, TMSBr may be involved in this part of the reaction, but this isn't shown in the mechanism. Otherwise, why would low valent Cr be required in a mechanism in which oxidation of *t*-BuMgBr is occurring?

We thank the Reviewer for the valuable comments and questions. TMSBr should play important role and be involved in this part of the reaction. We revised the related elementary step and shows the formation of (TMS)Cr(n+1) species in the revised manuscript. It may provide an explanation why low-valent Cr species is required in a mechanism in which oxidation of *t*-BuMgBr is occurring.

Other comments:

Is the formation of olefins observed when heavier Grignard reagents are used? This would also provide support for a beta-hydride elimination activation pathway.

Thank the Reviewer for the question. Indeed, we observed that small amount of related olefin was formed by GC-MS analysis when using Grignard reagent of (1-methylcyclohexyl)magnesium bromide in the alkylation (in case of forming product **38**).

Are saturation kinetics observed when excess benzamide is employed? This could indicate a pre-equilibrium with benzamide coordinating to the active chromium catalyst.

We thank the Reviewer for the valuable question. We studied the dependence of the initial reaction rate on *N*-methylbenzamide (**1**) with high concentrations. The reaction shows first-order kinetics for benzamide when the concentration was below ~ 0.19 M. It seems that the dependence of the reaction rate on the concentration of **1** was more complex with a high concentration (Please see the following Table for the related data). The reaction rate was significantly decreased when the concentration of **1** was higher than ~ 0.19 M. However, the alkylation almost cannot occur when increasing the concentration of **1** was up to 0.35 M. It's hard to say that the initial rate of benzamide exhibited a saturation behavior at high concentrations.

Table S5. Initial Rate Data Obtained by Varying the Concentration of 1

Entry	amide 1 [M]	^t BuMgBr 2 [M]	Initial rate [M/min]
1	0.0930	0.4761	4.41×10^{-4}
2	0.1163	0.4761	4.78×10^{-4}
3	0.1395	0.4761	5.81×10^{-4}
4	0.1628	0.4761	6.31×10^{-4}
5	0.1860	0.4761	6.83×10^{-4}
6	0.2326	0.4761	3.57×10^{-4}
7	0.3488	0.4761	0

In the description of conditions used in Figure 2, I think that “TMSBr (0.06 mmol)” should be “TMSBr (0.6 mmol).”

We thank the Reviewer for pointing out this error. It was corrected in the revised Figures 2 and 3.

“Interestingly, the para-alkylation with 10 mol % of FeCl₃, AlCl₃ or AlMe₃ also occurred, albeit with giving low conversions (entries 11 and 12)” should probably be “Interestingly, the para-alkylation with 10 mol % of FeCl₃, AlCl₃ or AlMe₃ also occurred, albeit with low conversions (entries 11 and 12).”

It has been corrected in the revised manuscript.

On page 10, the authors write: “The analysis of the alkylation of **1a** after 4 hours by EPR spectroscopy technique showed that radical intermediates are involved in the reaction.” EPR may show the presence of radical species, but this does not prove that they are actually involved in the

active catalytic cycle. I would re-phrase this sentence: “The analysis of the alkylation of 1a after 4 hours using EPR spectroscopy suggests the formation of radical species during this reaction.”

Thank the Reviewer for the kindly suggestion. This sentence was re-written accordingly.

Page 10: “singlet electron transfer” should be “single electron transfer.”

We thank the Reviewer for pointing out this mistake. It was corrected in the revised manuscript.

Are silylated products ever observed when *t*BuMgBr is used? The NMRs for several products (e.g., S85) suggest the possible presence of a silylated byproduct.

Thank the Reviewer for the questions. The silylated products were not observed when the use of *t*-BuMgBr in the reaction. The appeared signals around 0 ppm in NMR spectra of several products may attribute to silicon grease. Other signals in the ¹H and ¹³C NMR spectra are consistent with the desired products.

The Second Cycle of Response to the Editor and Reviewer's Comments

Response to the Editorial Requests:

Please provide a point-by-point response to these points with your submission.

* Nature Communications uses a transparent peer review system, where for manuscripts submitted from January 2016 we are publishing the reviewer comments to the authors and author rebuttal letters of our research articles online as a supplementary peer review file. Please let us know in the cover letter when submitting the final version of your manuscript if you wish to opt out of this scheme or not. If you are concerned about the release of confidential data, we do permit redactions in the interest of confidentiality. If you would like to remove such information from these files, then please let us know specifically what information you would like to have removed. Please note that we cannot incorporate redactions for other reasons. Reviewer names will be published in the peer review files if the reviewer comments to the authors are signed by the reviewer, or if reviewers explicitly agree to release their name. For more information, please refer to our FAQ page at:

<https://media.nature.com/full/nature-assets/ncomms/authors/ncomms-transparent-peer-review.pdf>

We thank the editor for the valuable information. A supplementary peer review including the reviewer comments and rebuttal letters will be submitted with the revised manuscript.

* Please ensure that an updated editorial policy checklist that verifies compliance with all required editorial policies is completed and uploaded with the revised article. All points on the policy checklist must be addressed; if needed, please revise your manuscript in response to these points. Please note that this form is a dynamic 'smart pdf' and must therefore be downloaded and completed in Adobe Reader.

Editorial policy checklist: <https://www.nature.com/authors/policies/Policy.pdf>

The updated editorial policy checklist has been completed and will be uploaded with the revised article. All points on the policy checklist have been addressed.

* Your manuscript should comply with our policies and format requirements, detailed in our checklist for authors at:

http://www.nature.com/article-assets/npg/ncomms/authors/ncomms_manuscript_checklist.pdf

We carefully check the format requirements, and the manuscript should comply with the policies and format requirements.

* Data availability statements and data citations policy: All Nature Communications manuscripts must include a section "Data Availability" at the end of the Methods section or main text (if no Methods). For more information on this policy, and a list of examples, please see <http://www.nature.com/authors/policies/data/data-availability-statements-data-citations.pdf> In particular, the Data availability statement should include:

- Accession codes for deposited data
- Other unique identifiers (such as DOIs and hyperlinks for any other datasets)
- At a minimum, a statement confirming that all relevant data are available from the authors
- If applicable, a statement regarding data available with restrictions
- If a dataset has a Digital Object Identifier (DOI) as its unique identifier, we strongly encourage

including this in the Reference list and citing the dataset in the Data Availability Statement.

The section “Data Availability” was added at the end of the Methods section in the revised manuscript.

* **DATA SOURCES:** We strongly encourage authors to deposit all new data associated with the paper in a persistent repository where they can be freely and enduringly accessed. We recommend submitting the data to discipline-specific, community-recognized repositories, where possible and a list of recommended repositories is provided

here: <http://www.nature.com/sdata/policies/repositories>

If a community resource is unavailable, data can be submitted to generalist repositories such as figshare (<https://figshare.com/>) or Dryad Digital Repository (<http://datadryad.org/>). Please provide a unique identifier for the data (for example a DOI or a permanent URL) in the data availability statement, if possible. If the repository does not provide identifiers, we encourage authors to supply the search terms that will return the data. For data that have been obtained from publically available sources, please provide a URL and the specific data product name in the data availability statement. Data with a DOI should be further cited in the methods reference section. Please refer to our data policies here: <http://www.nature.com/authors/policies/availability.html>

The X-ray crystallographic coordinates for structures that support the findings of this study have been deposited at the Cambridge Crystallographic Data Centre (CCDC) with the accession code CCDC 1821836 (3). All other data supporting the findings of this study are available within the article and Supplementary Information files.

* To ensure correct hyperlinking of the accession codes in your manuscript, please add the hyperlink or DOI in square brackets directly after the code throughout (for example, '5XRN [<http://dx.doi.org/10.2210/pdb5XRN/pdb>]', '1483958 [<https://dx.doi.org/10.5517/ccdc.csd.cc11t5m6>]', 'SRP109982 [<https://www.ncbi.nlm.nih.gov/sra/?term=SRP109982>]' or 'NQLW00000000 [https://www.ncbi.nlm.nih.gov/assembly/GCA_002312845.1/]').

The DOI should be added after the code throughout.

* Please check whether your manuscript or Supplementary Information contain third-party images, such as figures from the literature, stock photos, clip art or commercial satellite and map data. We strongly discourage the use or adaptation of previously published images, but if this is unavoidable, please request the necessary rights documentation to re-use such material from the relevant copyright holders and return this to us when you submit your revised manuscript.

The manuscript and Supplementary Information do not contain third-party images.

* Please supply the main manuscript file in Microsoft Word or LaTeX format.

The main manuscript file in Microsoft Word format should be supplied with the resubmission.

* The title must be less than 15 words and should not include punctuation. Please edit the title accordingly.

The manuscript title has been edited accordingly. Now the title is less than 15 words and not include punctuation.

* Please change "Results and Discussion" to "Results" and divide the Results and Methods section into subsections, each with a title of 60 characters or fewer including spaces. Please ensure that a subheading is present at the very beginning of both the Results and Methods sections, even if there is only one subsection.

"Results and Discussion" has been changed to "Results" in the revised manuscript, and the Results and Methods sections have been divided into subsections, each with a title of fewer including spaces. A subheading is present at the very beginning of both the Results and Methods sections.

* Please ensure all subheadings in the Results and Methods sections contain fewer than 60 characters including spaces.

We are sure that all subheadings in the Results and Methods sections containing fewer than 60 characters including spaces.

* Please remove phrases such as 'new', 'novel', 'for the first time', 'unprecedented', etc. as these are not needed to emphasise the importance of your work.

The manuscript does not contain phrases such as "new", "novel", "for the first time", "unprecedented", etc. as these are not needed to emphasise the importance of the work.

* We are committed to ensuring clarity and avoiding ambiguity in the mathematics in our papers. Consequently, please carefully check the mathematical terms throughout your manuscript and Supplementary Information (including labels on figures and figure captions) to ensure that it conforms strictly to the following guidelines. Equations should be supplied in editable format, and not as images. In mathematical terms, scalar variables (e.g. x , V , χ) should be typeset in italic, whereas multi-letter variables should be formatted without italic. Constants (e.g. \hbar , G , c) should be typeset in italics (the only exceptions being e , i , π , which should be typeset without italic) and vectors (such as r , the wavevector k , or the magnetic field vector B) should be typeset in bold without italics. In contrast, subscripts and superscripts should only be italicized if they too are variables or constants. Those that are labels (such as the 'c' in the critical temperature, T_c , the 'F' in the Fermi energy, E_F , or the 'crit' in the critical current, I_{crit}) should be typeset in roman. Please also ensure the same convention is followed in figure labels, axes, and such. Additionally, to avoid doubt, unit dimensions should be expressed using negative integers (e.g. $\text{kg m}^{-1} \text{s}^{-2}$ not kg/ms^2) or the word 'per'.

The mathematical terms throughout the manuscript and Supplementary Information have been carefully checked. Equations is supplied in editable format. The related ChemDraw (.cdx) files with the final version of the manuscript will be supplied with the revised manuscript.

* Please see our requirements regarding characterization of structurally-novel chemical compounds, and the required format for compound characterization data: <http://www.nature.com/ncomms/journal-policies/editorial-publishing-policies#Characterization-materials> Please note that this includes $^1\text{H-NMR}$, $^{13}\text{C-NMR}$ and high resolution mass spectrometry for all structurally-novel chemical compounds.

We thank the Reviewer for the valuable information.

* Chemical structures that appear in figures in the manuscript (or as part of the single .cdx file mentioned below), should be drawn using the Nature Chemistry template (available

at http://www.nature.com/authors/guides/NR_chemdraw_stylesheet.cds) or using the settings from this template. Structures should be scaled in proportion to fit our figure dimensions, however, please only scale atom labels and bond lengths; please do not reduce (or increase) the bond thickness. Please also refer to the Nature Research Chemical Structures Guide (<https://www.nature.com/authors/guides/ChemStructureGuide.pdf>) to ensure that you prepare your figures in a format that will require minimal changes by our Production teams. Please supply any ChemDraw (.cdx) files with the final version of your manuscript.

The related ChemDraw (.cdx) files with the final version of the manuscript will be uploaded.

* Please ensure that figure legend titles are brief - they should not occupy more than one line in the final proof.

The related figure legend titles have been edited accordingly. Now the figure legend titles are not occupy more than one line in the final proof.

* In figure 1, please change "prior art(s)" to "prior work".

In figure 1, "prior art(s)" was changed to "prior work".

* In order for us to accurately represent the data in your tables, they must conform to journal style; unless you format the tables in your manuscript as described here, they won't display correctly in the published paper. Data in tables must be free from bold/italic formatting unless this has been clearly defined in the footnote. We are also unable to process colour in tables so this should be removed. We cannot display tables that do not fit onto a single page or multi-element tables. Finally, we are unable to merge cells or include vertical dividing lines or diagonal lines.

The tables in the manuscript was corrected accordingly. Data in tables are free from bold/italic formatting, and the color in tables was removed. Meanwhile, the displaying tables fit onto a single page.

* Please make a 'Competing Interests' statement after the 'Author Contributions' section that refers to all authors. If there are no competing interests, please add the statement "The authors declare no competing interests."

The statement of "The authors declare no competing interests." was supplied after the "Author Contributions" section in the revised manuscript.

* Please note that we do not reformat Supplementary Information files; they will be uploaded with the published article as they are submitted with the final version of your manuscript. Please check everything very carefully and remove any track changes from the file. Failure to adhere to our style guidelines will result in delays in production. The only sections we permit in the Supplementary Information file are Supplementary Figures, Supplementary Tables, Supplementary Methods, Supplementary Notes, Supplementary Discussion, Supplementary References.

We checked the Supplementary Information file carefully and remove any track changes from the file. The related Figures, Table and References have been corrected as Supplementary Figure, Supplementary Table and Supplementary References in Supplementary Information file.

* In the Supplementary Information file, please ensure that supplementary items are labelled and

cited using only the following formats: Supplementary Figure 1, Supplementary Table 1, Supplementary Methods, Supplementary Note 1, Supplementary Discussion, Supplementary References. Please note the use of 'Supplementary' and that we do not use the 'S' prefix.

The Supplementary Information file was corrected accordingly. Now supplementary items are labelled and cited using the following formats: Supplementary Figure 1, Supplementary Table 1, Supplementary Methods, Supplementary References.

* Please replace general citations to the Supplementary Information (e.g. 'see Supplementary Information') with specific citations (e.g. 'See Supplementary Figure 1/Supplementary Table 1/Supplementary Methods/etc.').

General citations to the Supplementary Information were replaced with specific citations in the revised manuscript.

* Each Supplementary Figure should be accompanied by a legend, which should be presented below the figure and may be up to 350 words, that refers to all panels within the figure, and a title that summarises the figure and does not refer to specific panels. This also applies to spectra, which should be labelled as Supplementary Figures (currently they are not labelled).

The Supplementary Figures were corrected accordingly.

* Please ensure that the Supplementary References appear at the end of the SI, and are self-contained and numbered from 1. References mentioned in both the main text and the Supplementary Information should be part of both reference lists so that the Supplementary Information does not refer to the reference list in the main paper and vice versa.

The form of Supplementary References was corrected accordingly. They appear at the end of the SI and numbered from 1. The Supplementary Information does not refer to the reference list in the main paper.

* Please upload the "Cif Document for Product 3" as Supplementary Data 1 and supply legends for each Supplementary Data file in your cover letter (not in the Supplementary Information file).

The "Cif Document for Product 3" as Supplementary Data 1 was uploaded, and legends for each Supplementary Data file were supplied in the cover letter.

* Your paper will be accompanied by a two-sentence editor's summary, of between 250-300 characters, when it is published on our homepage. Could you please approve the draft summary below or provide us with a suitably edited version.

The selective installation of quaternary carbon substituents on arenes is a highly synthetic challenge. Here, the authors report a chromium-catalyzed *para*-selective functionalization of benzamides with tertiary organometallic reagents under mild conditions.

Thank the Editor for the valuable help.

We would like to edit a little bit about the version of draft summary as shown in following:

The site-specific installation of quaternary carbon substituents on arenes is a highly synthetic challenge. Here, the authors report a chromium-catalyzed *para*-selective functionalization of benzamides with tertiary organometallic reagents under mild conditions.

OPEN ACCESS:

Nature Communications is a fully open access journal. Articles are made freely accessible on publication under a CC BY license (Creative Commons Attribution 4.0 International License). This license allows maximum dissemination and re-use of open access materials and is preferred by many research funding bodies.

For further information about article processing charges, open access funding, and advice and support from Nature Research, please visit <http://www.nature.com/ncomms/about/open-access>

We thank the Editor for the valuable information.

SUBMISSION INFORMATION:

In order to accept your paper, we require the following:

- * A cover letter describing your response to our editorial requests.

The response to the editorial requests should be included in the cover letter.

- * A separate document detailing your point-by-point response to any issues raised by our referees (please include the referees' comments in this document).

A separate supplementary peer review detailing our point-by-point response to any issues raised by the referees will be submitted with the revised manuscript.

- * The final version of your text as a Word or TeX/LaTeX file, with any tables prepared using the Table menu in Word or the table environment in TeX/LaTeX and using the 'track changes' feature in Word.

The final version of the manuscript as a Word file with tables, and file of "track changes" will be uploaded.

- * Production-quality versions of all figures, supplied as separate files. To ensure the swift processing of your paper please provide the highest quality, vector format, versions of your images (.ai, .eps, .psd) where available. Please see our brief guide to manuscript submission for further details on the figure formats we can accept. Text and labelling should be in a separate layer to enable editing during the production process. If vector files are not available then please supply the figures in whichever format they were compiled in and not saved as flat .jpeg or .TIFF files. Any chemical structures or schemes contained within figures should additionally be supplied as separate ChemDraw (.cdx) files. If your artwork contains any photographic images, please ensure these are at least 300 dpi.

Production-quality versions of all figures were supplied as separate files with the form of png. The related chemical structures or schemes contained within figures were additionally be supplied as separate ChemDraw (.cdx) files.

To ensure that your figures are accessible to colour-blind readers, we encourage you to use

alternative colour schemes. For example, rainbow colour scales may be replaced by single-colour intensity scales or greyscale, and red/green image overlays may be replaced with magenta/green. For reference an example of R-script colour blindness palettes can be found here <https://cran.r-project.org/web/packages/viridis/vignettes/intro-to-viridis.html>. Another example for Python can be found here: <http://matplotlib.org/cmocan/>

The red image overlays were replaced with magenta in the revised manuscript.

* The final version of any Supplementary Information (figures, tables, notes etc) in one PDF file. Please add a cover page to the Supplementary Information PDF, including the title of the manuscript and the first author's surname in the format 'Smith et al.' Please submit movies, audio files and data sets as separate files.

See <http://www.nature.com/ncomms/submit/how-to-submit#Supplementary-information> for acceptable file formats/sizes.

** Please note that Supplementary Information must be finalised prior to acceptance of the paper.
**

The final version of the Supplementary Information in one PDF file with a cover page will be uploaded.

* If you wish, an interesting image (but not an illustration or schematic) for consideration as a 'Featured Image' on the Nature Communications homepage. Examples can be seen on our Facebook page: <http://go.nature.com/PGPizM> The file should be 1400x400 pixels in RGB format and should be uploaded as 'Related Manuscript File'. In addition to our home page, we may also use this image (with credit) in other journal-specific promotional material.

We thank the Editor for the valuable suggestion.

* A completed author checklist, uploaded as a Related Manuscript file type, available at: http://www.nature.com/article-assets/npg/ncomms/authors/ncomms_manuscript_checklist.pdf

A completed author checklist will be uploaded as a Related Manuscript file type.

* Completed and signed copies of our Multimedia License to Publish (LTP) for any Featured Image suggestions (please use one form for each image and give a scientific description of the image in the 'title' field; do not use "Featured Image" as a title):
Multimedia Licence to Publish form

At acceptance, the corresponding author will be required to complete an Open Access Licence to Publish on behalf of all authors, declare that all required third party permissions have been obtained and provide billing information in order to pay the article-processing charge (APC) via credit card or invoice.

Please note that your paper cannot be sent for typesetting to our production team until we have received these pieces of information; therefore, please ensure that you have this information ready when submitting the final version of your manuscript.

<http://mts-ncomms.nature.com/cgi-bin/main.plex?el=A7S4BGZE7B2FJcD2I4A9ftdsomJHi2GpY1mxTd5jRqSwZ>

We thank the Editor for the valuable information.

We thank the Editor for the valuable time and editorial efforts to process the manuscript.

Response to the Reviewer 1's comments

The revised manuscript now tries to address some of the issues raised by the reviewers. In particular, it was very important to address the role of TMSBr. In the revised manuscript, it is proposed to play a role in the formation of tBu radical. It is shown that TMS-[Cr] species is formed in the mechanistic scheme. This proposal may be supported by the observation that para-TMS-substituted benzamide is formed when iPr-MgBr is reacted. This result should be included in the main text because it can partially support the proposed mechanism. Also, the results with 1° and 2° alkyl Grignard reagents as well as other aromatic carbonyl compounds should also be mentioned in the main text. The authors should note that questions and comments from the reviewers should be regarded as representative of those that might be posed by readers of the article. Consequently, these questions and comments should be addressed not only in the response letter but also within the body of the resubmitted manuscript itself.

We thank the Reviewer for the valuable comments and suggestions.

The descriptions of “*However, the reaction between benzophenone and tert-butyl Grignard reagent did not afford the alkylated product. It was noted that primary and secondary alkylmagnesium bromide cannot react with N-methylbenzamide to give the alkylated compounds. Interestingly, the formation of para-TMS-substituted benzamide in low yield was observed when using isopropyl Grignard reagent in the alkylation.*” were supplied into the revised manuscript. Meanwhile, the comment of “*The observation of para-TMS-substituted benzamide compound when using i-PrMgBr may indicate that the formation of TMS-Cr intermediate could be considered.*” was putted into the mechanism discussion section.

With these revisions to improve the discussion about the mechanism in hand, the manuscript may warrant publication in Nature Communications.

We thank the Reviewer for the strong support.

Response to the Reviewer 3's comments

The revised manuscript has addressed the mechanistic questions that I originally had. The proposed mechanism is now more clearly depicted, and follow up mechanistic experiments have been conducted. I feel that this manuscript is now suitable for publication in Nature Communications.

We sincerely thank the Reviewer for the positive evaluation on this work.

One typo: In Figure 6, the y axes should be labeled "(initial)" not "(initiate)"

This error was corrected in the revised manuscript.